# Socially Inspired Coalition Formation and Client Selection in Federated Learning

## Abstract

Federated Learning (FL) enables privacy-preserving collaborative model training, but its effectiveness is often limited by client data heterogeneity. We introduce a client-selection algorithm that (i) dynamically forms non-overlapping coalitions of clients based on asymptotic agreement and (ii) selects one representative from each coalition to minimize the variance of model updates. Our approach is inspired by social-network modeling, leveraging homophily-based proximity matrices for spectral clustering and techniques for identifying the most the most informative individuals to estimate a group's aggregate opinion. We provide theoretical convergence guarantees for the algorithm under mild, standard FL assumptions. Finally, we validate our approach by benchmarking it against three strong heterogeneity-aware baselines; the results show higher accuracy and faster convergence, indicating that the framework is both theoretically grounded and effective in practice.

## 1 Introduction

Federated Learning (FL) enables collaborative model training across multiple clients without sharing raw, potentially sensitive data (McMahan et al., 2017). While FL reduces the need to centralize data and can improve privacy and regulatory compliance, its effectiveness is often limited by statistical heterogeneity. In practice, client data is typically non-IID, with each local dataset following a distinct distribution (Zhao et al., 2018; Li et al., 2020). This distribution shift leads to inconsistent local updates, slower convergence, and reduced generalization, undermining the benefits of collaboration. To address statistical heterogeneity, several approaches have been proposed. Regularization methods constrain local updates to remain close to the global model (Li et al., 2020; Li & Wang, 2019; Karimireddy et al., 2020b;a), while personalization techniques adapt models to individual clients or client groups (Smith et al., 2017; Finn et al., 2017). Clustering-based methods further mitigate heterogeneity by grouping clients with similar data distributions, enabling the training of specialized models or more informed aggregation (Ghosh et al., 2020; Sattler et al., 2020).

Concurrently, client-sampling strategies have gained prominence. In practical deployments, factors such as intermittent client availability, energy constraints, and limited bandwidth necessitate selecting only a subset of clients in each round (Bonawitz et al., 2019). While uniform random sampling is a common baseline, growing evidence suggests that non-uniform (biased) selection can accelerate convergence and improve model quality, especially under heterogeneity (Cho et al., 2022; Wang et al., 2020b; Goetz et al., 2019). Complementing these empirical advances, theoretical works have framed FL as a model-sharing game (Donahue & Kleinberg, 2021a;b), analyzing incentive compatibility and the Price of Anarchy under various client behaviors and aggregation strategies.

In this work, we introduce `FedCVR-Bolt` (Federated Coalition Variance Reduction with Boltzmann exploration), a client sampling algorithm that mitigates heterogeneity via two stages: (1) an adaptive coalition detection phase that dynamically groups clients with similar model states and (2) a within-coalition selection step using a biased Boltzmann-like probability measure to maximize the expected intra-coalition variance reduction (Gallavotti, 2013). Our primary focus is conceptual and methodological: we draw an analogy between client selection in FL and homophily-based social selection mechanisms Mäs et al. (2010), and we leverage results form estimation theory (Kay, 1993) to ground the variance–reduction principle at the core of `FedCVR-Bolt`.

**Paper structure** After a brief overview of related work, Section 2 introduces the FL framework and problem setup, drawing parallels with social and sensor networks that inspire our approach. A detailed presentation of the `FedCVR-Bolt` is given in Section 3, including a step-by-step description and the corresponding pseudocode. The core theoretical results that support the methodology are presented in Section 3.1, while proofs and additional theoretical considerations are deferred to the Appendix. In Section 5, we validate `FedCVR-Bolt` across a range of heterogeneous settings and common benchmarks in FL (Caldas et al., 2018), showing its capability to outperform existing baselines.

**Contribution** Our contributions are as follows: i) We propose `FedCVR-Bolt`, a client sampling algorithm, which improves performance in heterogeneous FL settings compared to existing sampling strategies. We draw on social-interaction models to introduce homophily-based proximity matrices for spectral clustering and to adapt most-informative-node selection for client choice. ii) We provide theoretical guarantees on the convergence of the `FedCVR-Bolt` algorithm under mild assumptions. iii) We introduce a synthetic dataset for benchmarking federated regression under controlled heterogeneity. iv) We empirically validate our approach on synthetic and real-world data, demonstrating strong performance under Non-IID conditions.

## RELATED WORK

**Federated Learning** Statistical heterogeneity across client data distributions can significantly impede the performance and convergence of the global model. Regularization methods, such as `FedProx` (Li et al., 2020), `FedMD` (Li & Wang, 2019), `Scaffold` (Karimireddy et al., 2020b), and `Mime` (Karimireddy et al., 2020a), constrain local client updates or modify the global aggregation process to improve robustness and convergence in heterogeneous settings. Personalization techniques adapt the global model or learn client-specific models tailored to individual data distributions, with notable examples including `Per-FedAvg` (Fallah et al., 2020) and `pFedMe` (T Dinh et al., 2020). Collaboration and clustering strategies identify and leverage similarities among clients by grouping them, enabling more effective federated training within these identified clusters; prominent approaches in this domain include `IFCA` (Ghosh et al., 2020), `CFL` (Sattler et al., 2020), `FeSEM` (Long et al., 2023). More recent studies on clustered FL show the possibility of forming groups via consensus-based optimization (Carrillo et al., 2024) or by training a pairwise discriminator to estimate similarities (Bao et al., 2023).

Client selection strategies prove to be an effective approach in FL for handling client heterogeneity. Various methods address data heterogeneity by prioritizing clients exhibiting higher local loss on the current global model estimate (Cho et al., 2022) or by aiming for diverse client participation (Balakrishnan et al., 2022). Other approaches focus on variability in clients in training processes (Diao et al., 2020), data and/or update quality (Liao et al., 2024), or energy efficiency (Li et al., 2019). Methods such as `FedCBS` (Zhang et al., 2023) prioritize clients to reduce sampled class imbalance. More advanced policies, such as `Oort` (Lai et al., 2021) and `Harmony` (Tian et al., 2022), employ analytical rules address the device-selection heterogeneity. A distinct research thread develops selection policies with reinforcement learning, typically framing client selection as a Markov decision process (Powell, 2021). Notable examples include `FedRank` (Tian et al., 2024), `AutoFL` (Kim & Wu, 2021), `Favor` (Wang et al., 2020a), and `FedMarl` (Zhang et al., 2022b).

**FL and Game Theory** Game theory provides a rigorous framework for modeling incentives and collaboration in FL. Early work in Guazzone et al. (2013) applied coalition formation games to energy-aware resource management in distributed systems, laying the groundwork for strategic collaboration, though not addressing FL's statistical challenges. In FL settings, Donahue & Kleinberg (2021a) used hedonic games to analyze coalition stability in federated linear regression, deriving MSE-optimal aggregation schemes, but limited their analysis to linear models. This was extended in Donahue & Kleinberg (2021b), which framed FedAvg as a coalition game, introducing the Price of Anarchy to quantify inefficiencies in naive collaboration. They proposed an optimal coalition formation algorithm of limited scalability, although submodularity of the cost function suggests potential for efficient approximations. Blum et al. (2021) studied equilibria in different FL collaboration structures (*one-for-one* vs. *all-for-all*), characterizing when each is optimal (under stylized utility models). To address dynamic interactions, Ota et al. (2022) proposed graphical coalitional games, forming coalitions via synergy measures such as cosine similarity or Improvement Classification Accuracy and

analyzing robustness under adversarial settings. For incentive-aligned client selection, Nagalapatti & Narayanam (2021) proposed `S-FedAvg`, using Shapley values to quantify client utility and exclude unhelpful participants, though computational cost limits scalability. Recently, Zhang et al. (2022a) introduced a framework to incentivize high-quality data contributions, directly tackling input-data integrity, but faced challenges in distributed quality assessment and efficiency.

## 2 PROBLEM FRAMEWORK

### 2.1 FEDERATED LEARNING PROBLEM

The standard FL (McMahan et al., 2017) involves $K$ clients (agents), each holding its own training data. Over $T$ communication rounds, the clients jointly estimate a global vector of $D$ parameters $\theta_{gl} \in \mathbb{R}^D$, often called the *global model*. Ideally, the estimate to be found minimizes the *global loss*:

$$\theta_{gl} \in \arg\min_{\theta \in \mathbb{R}^D} \mathcal{L}(\theta), \tag{1}$$

where $\mathcal{L}(\theta) := \sum_{k=1}^K \alpha_k \mathcal{L}_k(\theta)$ with $\mathcal{L}_k(\theta)$ denotes the client-level loss computed on client $k$'s local dataset. The weights $\alpha_k$ are chosen in a way that $\sum_{k=1}^K \alpha_k = 1$ with $\alpha_k \geq 0$, reflecting the relative importance of each client; they are often set proportional to the size of the client's local dataset, *i.e.,* $\alpha_k = n_k / \sum_{k=1}^K n_k$ where $n_k$ stands for the number of training samples of the $k$-th client.

**Weighted-average global model.** We henceforth focus on the simplified canonical setting in which the global model is just a weighted average of the client-side parameter vectors (local models). Starting from a random initialization of the global model, at each round $t \in [T]$, the server sends to the clients the current global model $\theta_{gl}(t)$. Each client then performs $S \in \mathbb{N}$ local training iterations of a stochastic optimizer (e.g., Stochastic Gradient Descent) to minimize their local loss $\mathcal{L}_k$, thus resulting in a locally updated model $\theta_k(t+1)$. Each client communicates the updated model back to the server, who aggregates the updates as a weighted average, namely

$$\theta_{gl}(t+1) = \sum_{k=1}^K \alpha_k \theta_k(t+1). \tag{2}$$

This setting is natural when all clients aim to estimate the same $D$-dimensional parameter vector using only their local datasets, under the assumption that all clients share the same model architecture.

**Two-stage updates.** The global model is updated once per communication round. With a slight abuse of notation, we denote by $\theta_{gl}(t)$ the global estimate produced at the end of round $t$. Because clients neither share raw data nor communicate directly with one another, the update from $\theta_{gl}(t)$ to $\theta_{gl}(t+1)$ follows a two-stage procedure: (i) the clients refine their local estimates $\theta_k(t)$ by training on their private data, initializing from the global parameters $\theta_{gl}(t)$ broadcast by the server; (ii) the central server then aggregates these client updates to obtain the next global model estimate $\theta_{gl}(t+1)$.

**Partial participation.** Since the number of clients $K$ can be very large, the server typically interacts with only a fraction of them in each round. This *partial-participation* strategy keeps the throughput and energy consumption manageable, mitigates stragglers, and thereby shortens each communication round without compromising convergence (McMahan et al., 2017; Luo et al., 2024). Namely, the global model is communicated to the randomly selected subset of clients $\mathcal{P}_t \subseteq \{1, \ldots, K\}$ whose size $P = |\mathcal{P}_t|$ is kept constant across rounds. Only these selected clients compute their local updates $\theta_k(t+1)$ and upload them to the server for aggregation. The server updates the global model by aggregating the received estimates:

$$\theta_{gl}(t+1) = \sum_{k \in \mathcal{P}_t} \tilde{\alpha}_k(t) \, \theta_k(t+1). \tag{3}$$

where the weights $\tilde{\alpha}_k$ satisfy $\tilde{\alpha}_k(t) \geq 0$ and $\sum_{k \in \mathcal{P}_t} \tilde{\alpha}_k(t) = 1$. These weights differ from the original $\alpha_k$ as they are re-normalized to the active subset $\mathcal{P}_t$.

**Statistical learning setup.** We now assume that each client's training set is a sample drawn from an underlying data-generating distribution. The parameter vector $\theta_k \in \mathbb{R}^D$ learned by client $k$ (usually, obtained by minimizing its local empirical risk) thus becomes a *random variable*. Consequently, the vector $\theta_k(t)$ produced by client $k$ in communication round $t$ is a realization of that random variable, which the server then incorporates into the global aggregation. We define the random vector $\theta^d = (\theta_1^d, \ldots, \theta_K^d)^\top \in \mathbb{R}^K$ whose entries are the $d$-th components of the clients' parameter

vectors, denoting its mean by $\bar{\theta}^d = \mathbb{E}(\theta^d)$ and its covariance matrix by $C^d = \mathbb{C}ov(\theta^d) \in \mathbb{R}^{K \times K}$. Entries of $C^d$ quantify client *heterogeneity* for the $d$-th model parameter: its diagonal elements capture the variance across clients, and the off-diagonal entries measure pairwise similarity between clients' values. Without loss of generality, we will assume that $C^d$ is an invertible matrix. By Equation equation 2, the components of the global model are scalar random variables $\theta_{gl}^d = \alpha^\top \theta^d$.

## 2.2 PROBLEM SETUP AND SOCIAL DYNAMICS ANALOGY

Recent studies show that, under data heterogeneity, the choice of the participating-client set $\mathcal{P}_t$ is critical for convergence as *biased* sampling policies can significantly accelerate and stabilize training (Ghosh et al., 2020; Cho et al., 2022). Building on this, we introduce a FL algorithm inspired by social-dynamics models that sample clients via a two-step procedure. First, we form coalitions whose covariance mirrors client relationships, explicitly encouraging high intra-cluster and low inter-cluster covariance, so that each cluster captures a distinct sub-distribution. Second, within each cluster, we sample clients to train the global model on representatives that best preserve this structure, thereby directly mitigating the effect of heterogeneity. This address two key challenges: i) how to partition the client data distributions; ii) how to select the clients to sample within each cluster.

Specifically, as explained in Appendix B, in our study we consider each client as a node (or agent) in a time-varying network. The set of agents $\mathcal{K} = \{1, \ldots, K\}$ refers to the set of clients of the federations and their "opinion" corresponds to the individual client model $\theta_k$. The influence matrix $W(t)$ encodes the pairwise influence between clients at time $t$, capturing phenomena such as homophily (Mäs et al., 2010), *i.e.,* agents are mainly influenced by individuals who hold similar opinions. This modeling framework is fundamental to identify coalitions across the network, a key challenge studied both in Clustered FL (Sattler et al., 2020) and in social community detection (Mark, 2003; Mäs et al., 2010). In practical scenarios, it is common to find groups of clients with similar data distributions and, consequently, similar local models. Such clients are naturally inclined to collaborate, so identifying coalitions is central to effective FL design, aligned with well established methods like Clustered FL (Sattler et al., 2020), IFCA (Ghosh et al., 2020) and FeSEM (Long et al., 2023). Accordingly, we partition the federation into $P$ groups of cooperating clients whose data are drawn from the same (or closely related) distribution, and then, at each round, select the *best* representative from each group using the strategy detailed below.

A common challenge, originally arising in the context of sensor placement problems (Das & Kempe, 2008; 2018) and later extended to social networks (Raineri et al., 2025; 2023) is the estimation of the overall network opinion when only a subset can be observed. Analogously, here, our goal is to select the optimal subset of clients to sample to best approximate the global model $\theta_{gl}$. Based on the cited literature approach and building on the statistical learning setup previously introduce, we will introduce a novel sampling strategy based on variance reduction techniques.

## 3 FEDCVR−BOLT ALGORITHM DESIGN

**Analogy with Opinion Dynamics** First, let us formalize the underlying network structure aimed to study. Let $\theta(t)$ indicate the vector of current models at round $t$, such that $\theta_k(t)$ is the model associated to client $k$ at communication round $t$. As introduced in (Mäs et al., 2010) a natural way to implement **homophily** it to define social influence of agent $j$ on agent $k$ as

$$W_{kj}(t) = \frac{e^{-\gamma \|\theta_k(t) - \theta_j(t)\|_2^2}}{\sum_{j \in \mathcal{K}} e^{-\gamma \|\theta_k(t) - \theta_j(t)\|_2^2}} \tag{4}$$

where $\gamma > 0$ is a parameter which amplifies the role of models similarities, *i.e.,* the larger $\gamma$, the more pronounced is the homophily effect. The resulting matrix admits a clear social-systems interpretation: agents (clients) with similar "opinions" (model parameters) exert stronger mutual influence and naturally form clusters. Importantly, computing this matrix, which is standard practice in clustered and personalized FL (Sattler et al., 2020; Bao et al., 2023), here reduces to evaluating L2 distances between server-available parameter vectors. To further validate our approach, we additionally implemented our algorithm with other similarity functions used in coalition formation (Scholkopf & Smola, 2001); see Appendix B.1.

**Coalition Formation**    Move now the focus on the coalitions formed which may vary with respect to the previous round. Formally, the goal of this step is to identify the $P$ coalitions formed at round $t$ that will be indicated by $\mathcal{C}_1, \ldots, \mathcal{C}_P \subset \Pi(K)$, where $\Pi(K)$ denotes the set of all possible partitions of $[K]$. Notice that the number of communities $P$ neither is assumed to be the true number of clusters in the data, nor is an output of the clustering step, but it is fixed a priori and it coincides with the model participation rate. This assumption is not restrictive since it is a common practice in clustered framework (e.g., refer to IFCA (Ghosh et al., 2020) and FeSEM (Long et al., 2023)).

As proposed in Narantsatsralt & Kang (2017), a natural way to detect communities in social networks is spectral clustering. The idea is now to project the data, collected in matrix $W$, in the corresponding eigenvector space and then to properly apply a k-means on this space. The projection on eigenvectors space is crucial since it allows to significantly distinguish the similar nodes into more distanced positions in feature space, assigning similar values to members of the same community and capturing also soft boundaries between communities, both convex and non-convex.

**Sampling Strategy**    Once the $P$ coalitions of clients have been identified, for each coalition our goal is now to identify the optimal client to sample at round $t$. Let us introduce the value vectors $v^d \in \mathbb{R}^K$ for the $d$-th model component, where each entry is defined as

$$v_k^d = \frac{(C^d\alpha)_k^2}{C_{kk}^d} \quad \text{for all } k \in \mathcal{K}. \tag{5}$$

As it will be explained in Section 3.1, $v_k^d$ refers to the variance reduction for the $d$-th component of global model $\theta_{gl}^d$ estimation subjected to $\theta_k^d$ sampling (Raineri et al., 2025).

Once computed the vector $v^d$ for any $d = 1, \ldots, D$, let us now introduce the vector $v \in \mathbb{R}^D$, s.t. $v_k$ is a global measure related to variance reduction on model $\theta_{gl}$ subjected to sample $\theta_k$. Precisely, we choose as a collective measure the total variance reduction defined as

$$v_k := \sum_{d=1}^{D} v_k^d, \tag{6}$$

which is a natural choice (Johnson & Wichern, 2002) for the overall residual variance for the global model $\theta_{gl}$, see Section 3.1 for theoretical details.

Within the defined setting, we select for each coalition the client maximizing vector $v$, *i.e.*, for each coalition $p = 1, \ldots, P$ we select the client $j_p$ such that

$$j_p \in \arg\max_{k \in \mathcal{C}_p(t)} v_k. \tag{7}$$

Selecting one representative per cluster mitigates overfitting to any single sub-distribution while preserving global diversity. Optimizing for variance reduction captures cross-client coupling, yielding more stable updates.

In real case scenario, the optimization in equation 7 becomes overly restrictive due to the unavailability of the true data distribution, which introduces an inherent estimation error in the covariance matrix. To mitigate the impact of this uncertainty while still exploiting the informative content of the estimated variance, we incorporate an additional exploration term into the objective.

**Boltzmann Exploration**    Instead of a greedy, deterministic selection based on the estimated variance, we employ a Boltzmann exploration policy (Gallavotti, 2013). This introduces a controlled stochasticity that helps mitigate the risk of becoming trapped by early biased or noisy estimates of the covariance matrix (Powell, 2022). Drawing an analogy with statistical mechanics (Gallavotti, 2013), where the probability of a system occupying a specific energy state is proportional to the exponential function of the negative energy divided by temperature, we let the probability of selecting a client $k$ within coalition $\mathcal{C}_p(t)$ during round $t$ based on its associated value $v_k$ be an indicator of the desirability of sampling client $k$. Higher values indicate greater potential to reduce the variance of the global model estimate. This Boltzmann-like probability measure is formally defined as:

$$\pi_p(k; t) = \frac{e^{\beta v_k}}{\sum_{j \in \mathcal{C}_p(t)} e^{\beta v_j}} \tag{8}$$

where $\pi_p(k; t)$ is the probability that client $k$ in coalition $\mathcal{C}_p(t)$ is selected at round $t$. The parameter $\beta$, analogous to the inverse temperature in statistical mechanics, controls the level of exploration. In

---

**Algorithm 1** FedCVR-Bolt Algorithm

---

1: **Inputs:** $K, D, T, \{\theta_k(1)\}_{k=1}^{K} \subset \mathbb{R}^D, \{C^d(1)\}_{d=1}^{D} \subset \mathbb{R}^{D \times D}, \theta_{gl}(1), \gamma_t, P$
2: **for** $t = 1, \ldots, T-1$ **do**
3:     Normalize: $\tilde{\theta}_k(t) = \theta_k(t)/\|\theta_k(t)\|$
4:     Compute similarities: $\rho_{kj} = \langle \tilde{\theta}_k(t), \tilde{\theta}_j(t) \rangle$
5:     Cluster clients: $\mathcal{C}_1(t), \ldots, \mathcal{C}_P(t) \leftarrow \text{SpectralClustering}(\tilde{\theta}_1(t), \ldots, \tilde{\theta}_K(t))$
6:     Compute variances $v_k^d$ with $C^d(t)$; set $v_k = \sum_{d=1}^{D} v_k^d$
7:     Compute Boltzmann probabilities: $\pi_p(k,t) = e^{v_k}/\sum_{j \in \mathcal{C}_p(t)} e^{v_j}$
8:     Sample $j_p \sim \pi_p(\cdot, t)$ in each cluster $\mathcal{C}_p(t)$; set $\mathcal{P}_t = \{j_1, \ldots, j_P\}$
9:     **for all** $j_p \in \mathcal{P}_t$ **do**
10:         Receive $\theta_{gl}(t)$, perform local update $\rightarrow \theta_{j_p}(t+1)$
11:     **end for**
12:     **for all** $p = 1, \ldots, P$ and $k \in \mathcal{C}_p(t)$ **do**
13:         $\bar{\theta}_k(t+1) = \rho_{kj_p}\theta_{j_p}(t+1)$
14:     **end for**
15:     **for** $d = 1, \ldots, D$ **do**
16:         $C^d(t+1) = (1-\gamma_t)C^d(t) + \gamma_t \left(\theta^d(t+1) - \bar{\theta}^d(t+1)\right)\left(\theta^d(t+1) - \bar{\theta}^d(t+1)\right)^\top$
17:     **end for**
18:     $\theta_{gl}(t+1) = \sum_{p=1}^{P} \tilde{\alpha}_{j_p}\theta_{j_p}(t+1)$     with $\tilde{\alpha}_{j_p} = n_{j_p}/\sum_{p=1}^{P} n_{j_p}$
19: **end for**

---

our experiments, we set $\beta = 1$. This design assigns clients with larger estimated variance-reduction $v_k$ proportionally higher selection probabilities, aligning sampling with their expected contribution within each coalition. Moreover, this framework keeps nonzero probability for lower-scoring clients, preventing myopic exploitation and enabling gains under uncertainty.

**Update Policy** Once the participating clients $\mathcal{P}_t = \{j_1, \ldots, j_P\}$ have been selected, the server sends them the current model. Then, they locally updates the central model according to their local dataset, obtaining the individual update $\theta_{j_p}(t+1)$ which is sent to the server. Finally, it coherently updates the global model as $\theta_{gl}(t+1) = \sum_{p=1}^{P} \tilde{\alpha}_{j_p}\theta_{j_p}(t+1)$.

Let us specify that for all the unobserved clients $k \in [K] \backslash \mathcal{C}_p(t)$ the value at round $t+1$ is considered equal to the previous one at round $t$. Formally, $\theta_k(t+1) = \theta_{j_p}(t+1)$ if $k = j_p$, and $\theta_k(t+1) = \theta_k(t)$ if $k \neq j_p$. The expected value $\bar{\theta}_k(t+1)$ of the unobserved clients $k \in [K] \backslash \mathcal{C}_p(t)$ is updated based on the observation of the sampled client $j_p$ *i.e.*,

$$\bar{\theta}_k(t+1) = \rho_{kj_p}\theta_{j_p}(t+1), \tag{9}$$

with $\rho_{kj_p}$ the Pearson correlation coefficient between models of client $k$ and $j_p$. As explained in Section 3.1, this is a natural choice as it coincides with taking the expected value of $\theta_k$ conditioned to observation $\theta_{j_p}$, *i.e.*, $\mathbb{E}[\theta_k | \theta_{j_p}(t+1)]$. Precisely, the rescaling parameter coincides with the optimal regression coefficient of $\theta_k$ given $\theta_{j_p}(t+1)$ under the assumption of normalized models.

**Online estimation of covariance** A central core of our method is to estimate the covariance matrices $C^d$ of the individual model's components. Let us denote by $C^d(t)$ the current estimate at round $t$. Then, the following formulation holds

$$C_{kj}^d = \mathbb{C}ov(\theta_k^d, \theta_j^d) = \mathbb{E}[(\theta_k^d - \bar{\theta}_k^d)(\theta_j^d - \bar{\theta}_j^d)]. \tag{10}$$

Using the $k, j$-th entry of the covariance matrix in equation 10, it is possible to obtain an online estimate of the covariance matrix, using the Robbins-Monro estimation (Robbins & Monro, 1951), as

$$C^d(t+1) = (1-\gamma_t)C^d(t) + \gamma_t[\theta^d(t+1) - \bar{\theta}^d(t+1)][\theta^d(t+1) - \bar{\theta}^d(t+1)]^\top \tag{11}$$

with $\bar{\theta}^d(t+1)$ the mean vector updated according to equation 9.

### 3.1 THEORETICAL DERIVATION OF FEDCVR-BOLT

This section presents the core theoretical results that underpin and justify each step of the proposed algorithm, providing the formal grounding for our methodology. For complete proofs and additional theoretical insights, we refer the reader to Appendix B. Let us introduce the following notation, which will be utilized subsequently. Given a matrix $X \in \mathbb{R}^{K \times K}$, and given $\mathcal{A}, \mathcal{B} \subseteq [K]$, we indicate with $X_{\mathcal{A}\mathcal{B}}$ the submatrix of $X$ having rows in $\mathcal{A}$ and columns in $\mathcal{B}$.

**Sampling Strategy**  Finding the optimal subset of clients to sample in order to obtain the best estimate of the global model is an instance of the more general and widely studied subset selection problem which is known to be NP-complete (Das & Kempe, 2008). Generalizing Raineri et al. (2025), let us here reformulate the main results on the specific context of interest. The first step consists in defining the proper evaluation metric. Specifically, we focus on Variance Reduction which measures the reduction in uncertainty on the variable to estimate, *i.e.*, $\theta_{gl}$, conditioned to the observation done. Formally, given an arbitrary subset of clients $\mathcal{A} \subseteq \mathcal{K}$, for each model component $d$, we denote the variance reduction on $\theta_{gl}^d$ conditioned to the observations $\theta_{\mathcal{A}}^d$ as

$$v_{\mathcal{A}}^d := \mathbb{V}ar(\theta_{gl}^d) - \mathbb{E}[\mathbb{V}ar(\theta_{gl}^d|\theta_{\mathcal{A}}^d)]. \tag{12}$$

First, let us introduce a preliminary Lemma, which comes from Lemma 2 in Raineri et al. (2025), which finds an explicit formulation for the global model expectation conditioned to samples $\mathcal{A} \subseteq \mathcal{K}$.

**Lemma 1.** *Consider $\theta^d \in \mathbb{R}^K$ the vector containing all the $d$-th model components of the federation and $\theta_{gl}^d = \alpha^\top \theta^d$ the global model to estimate. Given $\mathcal{A} \subseteq K$, it holds[1] $\mathbb{E}[\theta_{gl}^d|\theta_{\mathcal{A}}^d] = (C_{\mathcal{A}\mathcal{A}}^d)^{-1}(C^d\alpha)_{\mathcal{A}}\theta_{\mathcal{A}}^d$, where $C_{\mathcal{A}\mathcal{A}}^d$ is the invertible covariance matrix of $\theta_{\mathcal{A}}^d$.*

This lemma is crucial, as shown in Appendix B, provide an explicit formulation for variance reduction function in equation 12, properly generalizing Proposition 2 in Raineri et al. (2025).

**Proposition 1.** *Consider $\theta^d \in \mathbb{R}^K$ the vector containing all the $d$-th model components of the federation and $\theta_{gl}^d = \alpha^\top \theta^d$ the global model to estimate. Then, the variance reduction, subjected to sampling the subset $\mathcal{A}$ of possible clients, is computed as $v_{\mathcal{A}}^d = (C^d\alpha)_{\mathcal{A}}^\top (C_{\mathcal{A}\mathcal{A}}^d)^{-1}(C^d\alpha)_{\mathcal{A}}$, where $C^d$ is the covariance matrix which captured the correlations among the clients.*

Let us notice that if we consider a sample made of just one client, *i.e.*, $\mathcal{A} = \{k\}$, the previous results take a simpler and less computational expensive formula. This will be a core element in our sampling strategy since it will be used to evaluate the target function of interest.

**Corollary 1.** *Consider $\theta^d \in \mathbb{R}^K$ the vector containing all the $d$-th model components of the federation and $\theta_{gl}^d = \alpha^\top \theta^d$ the global model to estimate. Let $\theta_k^d$ be the sampled model at round $t$. Then, the variance reduction for the global model $\theta_{gl}^d$ given sample $\theta_k^d$ is $v_k^d = \frac{(C^d\alpha)^2}{C_{kk}^d}$.*

So far, we succeed in finding an explicit formulation for the variance reduction function of the global model $\theta_{gl}^d$ for each $d$-th component. Let us now define a collective measure of the overall variance reduction of $\theta_{gl}$. Based on Johnson & Wichern (2002), we define the *total variance reduction* as the sum of variance reductions associated to the different model components, *i.e.*, $v_k = \sum_{d=1}^{D} v_k^d$, which is the trace of $\mathbb{V}ar(\mathbb{E}(\theta_{gl}|\theta_k))$, coinciding with the sum of the $D$ squared deviation vectors.

**Update Policy**  Once the optimal model to sample is selected through the sampling strategy, we use this additional information to update all the other models distribution. Preliminary, let us recall the definition of the *Pearson Correlation Coefficient*, *i.e.*, given two random variables $X_i$ and $X_j$ out of a random vector $X$ with covariance matrix $C$, it holds $\rho_{jk} = \frac{C_{kj}}{\sqrt{C_{kk}}\sqrt{C_{jj}}}$. We can now define the update policy for the $j$-th model given the latest $k$-th sample. For more details see Appendix B.

**Proposition 2.** *Let $\theta^d \in \mathbb{R}^K$ be the vector containing all the $d$-th model components of the federation and $\theta_{gl}^d = \alpha^\top \theta^d$ the global model to estimate. Let $\theta_k^d$ be the sampled model at round $t$. Then, the optimal estimator of an arbitrary model $d$-th component $\theta_j^d$ with $j \neq k$ is $\mathbb{E}[\theta_j^d|\theta_k^d] = \rho_{kj}\frac{\sqrt{v_j}}{\sqrt{v_k}}\theta_k^d$.*

---

[1]Here, with some abuse of notation, $\mathbb{E}[\theta_{gl}^d|\theta_{\mathcal{A}}^d]$ denotes the projection of $\theta_{gl}^d$ onto the space of all linear combinations of the components of $\theta_{\mathcal{A}}^d$ with constant coefficients, found via linear regression. Under standard assumption of normal distribution it would coincide with the true conditional expectation (Hastie et al., 2009).

**Corollary 2.** *Consider $\tilde{\theta}^d \in \mathbb{R}^K$ the normalized vector containing all the $d$-th model components of the federation, such that $\mathbb{V}ar(\tilde{\theta}^d) = 1$. Let $\tilde{\theta}_k^d$ be the sampled model at round $t$. Then, the optimal estimator of an arbitrary model $d$-th component $\tilde{\theta}_j^d$ with $j \neq k$ is $\mathbb{E}[\tilde{\theta}_j^d | \tilde{\theta}_k^d] = \rho_{kj} \tilde{\theta}_k^d$.*

## 4 ALGORITHM PERFORMANCE ANALYSIS

**Algorithm Convergence Analysis**   At each step every selected client $k$ performs local training iterations of a stochastic optimizer (e.g., Stochastic Gradient Descent) which ensures non-increasing local loss $\mathcal{L}_k$. Then, since the global loss is a linear combination of the local ones, i.e., $\mathcal{L}(\theta) := \sum_{k=1}^K \alpha_k \mathcal{L}_k(\theta)$, it is non-increasing. Our method can be seen as an instance of partial sampling strategy which is known to preserve the optimality guarantees. Details are provided in Appendix B.4

**Computational Cost Computation**   The main overhead lies in performing spectral clustering on the server at each communication round. The computational cost at each round can be broken down as follows (von Luxburg, 2007). First, for the similarity matrix construction, pairwise similarities between the $K$ clients are computed using a RBF kernel. This requires evaluating squared Euclidean distances between $K$ vectors in $\mathbb{R}^D$, resulting in a complexity of $\mathcal{O}(K^2 D)$. Second, constructing the normalized Laplacian from the similarity matrix incurs a cost of $\mathcal{O}(K^2)$. Third, extracting the top $P$ eigenvectors of the $K \times K$ Laplacian requires $\mathcal{O}(K^2 P)$ operations in standard implementations. Fourth, the final step clusters the $K$ clients in the $P$-dimensional eigenspace, with cost $\mathcal{O}(KPI)$, where $I$ is the number of K-means iterations (in our setting constant set to 300). Overall, the dominant cost is $\mathcal{O}(K^2 D + K^2 P)$. In practice, since $P \ll D$ (e.g., $P = 10$, $D = 300$), the complexity is effectively driven by the similarity matrix computation, yielding a leading cost of $\mathcal{O}(K^2 D)$ per round. This overhead is modest and entirely server-side, since server resources are not a bottleneck.

## 5 EXPERIMENTS: FEDCVR-BOLT IN HETEROGENEOUS SCENARIOS

We present experimental results evaluating `FedCVR-Bolt` in heterogeneous FL scenarios. We employ established FL benchmark datasets (Caldas et al., 2018; Li et al., 2020), together with a synthetic dataset specifically designed for a controlled federated linear regression analysis. A detailed description of dataset settings and model architectures is provided in Appendix C.

In our analysis, we compare `FedCVR-Bolt` against a diverse set of FL baselines. From client selection methods, we include the uniform random policy of `FedAvg` (McMahan et al., 2017), the exploitative `Power-of-Choice` strategy (Cho et al., 2022) that favors clients with higher local losses, and `ActiveFL` (Goetz et al., 2019), which balances exploration and exploitation through probabilistic selection. From the broader FL literature, we also consider `FedProx` (Li et al., 2020), a regularization-based approach designed for heterogeneous settings, and two clustering-based personalized methods, `IFCA` (Ghosh et al., 2020) and `FeSEM` (Long et al., 2023). Section 5.1 reports regression results, highlighting the behavior of selection policies in controlled heterogeneous environments. Section 5.2 then turns to classification tasks on real-world datasets.

### 5.1 FEDERATED LINEAR REGRESSION

We simulate heterogeneity using $J \in \{1, 2\}$ clusters across $K = 100$ clients. Each client $k$ belongs to a cluster $j_k$, and their local inputs $x_k^i \in \mathbb{R}^D$ are sampled as $x_k^i \sim \mathcal{D}(\theta_{x,j_k}, \sigma_{x,j_k}^2 I_D)$. For each sample, a latent parameter $\theta_k^i \sim \mathcal{D}(\bar{\theta}_{j_k}, \sigma_{\theta,j_k}^2 I_D)$ is drawn, and the label is computed as $y_k^i = (\theta_k^i)^\top x_k^i$.

Cluster parameters, client assignments, and train/test splits follow uniform sampling routines, detailed in Appendix C. Table 1 reports the average test MSE after $T = 100$ rounds, selecting $P = 10$ clients per round with $S = 10$ local SGD steps. In IID settings ($J = 1$), all policies perform similarly, with `FedCVR-Bolt` maintaining competitive results. In non-IID scenarios ($J = 2$), `FedCVR-Bolt` consistently outperforms baselines. For instance, it reduces test MSE by up to $3.0\%$ over uniform sampling in the no-intercept, $D = 1$ case, and by $1.7\%$ when an intercept is included.

Table 1: Comparison between `FedCVR-Bolt` and FL selection baselines on a synthetic regression dataset. We evaluate two cases: $D = 1$ (linear model without intercept) and $D = 2$ (linear model with intercept), under both IID and non-IID settings. We report the Test MSE, averaged across clients (the lower the better). In the IID case, all methods perform comparably. In the non-IID case, `FedCVR-Bolt` achieves the lowest MSE, improving performance by approximately 3% for $D = 1$ and 1.7% for $D = 2$.

| Regression Model | Heterogeneity | FedAvg | Power-of-Choice | AFL | FedCVR-Bolt |
|---|---|---|---|---|---|
| $y = \theta_1 x$ | IID | $6.0171 \pm 2.4884$ | $6.3587 \pm 2.0280$ | $6.1656 \pm 2.5821$ | $\mathbf{6.0136} \pm \mathbf{2.5170}$ |
| | Non-IID | $57.1336 \pm 24.1335$ | $56.9701 \pm 24.7835$ | $56.8328 \pm 25.7835$ | $\mathbf{54.7102} \pm \mathbf{24.8973}$ |
| $y = \theta_0 + \theta_1 x$ | IID | $0.2142 \pm 0.3520$ | $0.2143 \pm 0.3533$ | $0.2134 \pm 0.3503$ | $\mathbf{0.2085} \pm \mathbf{0.3386}$ |
| | Non-IID | $1.5875 \pm 0.7291$ | $1.6742 \pm 0.6820$ | $1.7023 \pm 0.7642$ | $\mathbf{1.5127} \pm \mathbf{0.7336}$ |

Table 2: Comparison with FL baselines on heterogeneous classification benchmarks. We report the test accuracy (Hossin & Sulaiman, 2015), averaged across clients (the larger the better). `FedCVR-Bolt` consistently outperforms existing methods across MNIST, CIFAR-10, and CIFAR-100, all partitioned with Dirichlet $\alpha = 0.1$.

| Dataset | FedAvg | Power-of-Choice | Active FL | FedProx | IFCA | FeSEM | FedCVR-Bolt |
|---|---|---|---|---|---|---|---|
| MNIST | $86.30 \pm 1.12$ | $81.00 \pm 2.10$ | $88.89 \pm 0.98$ | $89.69 \pm 0.91$ | $89.46 \pm 1.04$ | $86.77 \pm 1.85$ | $\mathbf{90.23} \pm \mathbf{0.82}$ |
| CIFAR-10 | $52.67 \pm 1.25$ | $49.85 \pm 1.71$ | $55.19 \pm 1.34$ | $46.44 \pm 1.88$ | $52.94 \pm 1.09$ | $44.66 \pm 2.06$ | $\mathbf{57.06} \pm \mathbf{0.77}$ |
| CIFAR-100 | $22.92 \pm 0.95$ | $23.46 \pm 1.05$ | $23.35 \pm 1.12$ | $22.13 \pm 1.34$ | $22.55 \pm 1.28$ | $11.87 \pm 0.77$ | $\mathbf{24.82} \pm \mathbf{0.88}$ |

## 5.2 FL CLASSIFICATION BENCHMARKS

We next evaluate `FedCVR-Bolt` on standard FL classification benchmarks. We consider MNIST (LeCun, 1998), CIFAR-10, and CIFAR-100 (Krizhevsky, 2009), each partitioned across clients using a Dirichlet distribution with concentration parameter $\alpha = 0.1$ (Caldas et al., 2018), which introduces both label and quantity heterogeneity. Each dataset is trained for $T = 100$ communication rounds with $P = 10$ clients sampled per round and $S = 10$ local epochs. Model architectures and hyper-parameters are detailed in Appendix C.

Table 2 reports average test accuracy across clients. `FedCVR-Bolt` consistently achieves the highest performance across all benchmarks. In particular, it delivers notable gains over random client sampling (`FedAvg`) and over the loss-aware `Power-of-Choice`, which can fail to generalize well in heterogeneous regimes. Regularization with `FedProx` and clustering-based methods such as `IFCA` and `FeSEM` yield competitive results, but remain below `FedCVR-Bolt`. This confirms the effectiveness of our selection strategy in both simple and highly heterogeneous datasets.

## 6 CONCLUSIONS

We introduce `FedCVR-Bolt`, a novel framework addressing heterogeneous FL by drawing foundational insights from opinion dynamics models. The core of `FedCVR-Bolt` lies in a client selection strategy that biases sampling towards clients whose local models contribute most significantly to maximizing the variance reduction of the global model update. Distinct from conventional sampling methods, `FedCVR-Bolt` exploits concepts from coalition formation within opinion dynamics to identify clusters of clients exhibiting similar model characteristics, i.e., the "opinions". A representative client is then sampled from each identified cluster. This approach is supported by theoretical results that provide a strong motivation and foundation for our algorithm. Empirical evaluations on heterogeneous datasets, including a synthetic linear regression task specifically designed to highlight heterogeneity challenges and more complex benchmarks, demonstrate that `FedCVR-Bolt` consistently outperforms traditional FL client selection algorithms.

# 7 REPRODUCIBILITY STATEMENT

The reproducibility of the results and the theoretical contributions of this work has been a paramount concern throughout the entire development of this project and while drafting this manuscript. In this section, we provide details and a concise guide to reproduce our results and verify our contributions.

- **Code Availability:** All the code used in our experiments has been included in the supplementary material of the submission. Additionally, we will release online a well-documented and structured final version of the code, to allow for easy reproduction of the experiments detailed in the paper.

- **Datasets and data splits:** The datasets used in our experiments are publicly available and can be downloaded online. Detailed instructions on how to access and preprocess the datasets will be provided together with the final code release. The data splits can be generated directly through our provided code. This ensures that others can replicate the exact federated learning scenarios used in our work.

- **Theoretical Results:** We provide complete proofs for all theorems and propositions presented in the paper in Appendix B.

We are confident that with these resources, all experimental and theoretical results can be reproduced by the community.

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

| Symbol | Description |
|---|---|
| $K$ | number of clients |
| $\mathcal{K}$ | set of clients |
| $\mathcal{A} \subseteq \mathcal{K}$ | subset of clients |
| $W(t)$ | influence matrix among clients at round $t$ |
| $T$ | number of communication rounds |
| $D$ | number of model parameters |
| $S$ | number of training iterations |
| $P$ | number of participating clients |
| $\eta$ | learning rate |
| $\theta_{\mathrm{gl}} \in \mathbb{R}^D$ | global model |
| $\theta_{\mathrm{gl}}(t) \in \mathbb{R}^D$ | global model at round $t$ |
| $\theta_k(t) \in \mathbb{R}^D$ | local model of client $k$ at round $t$ |
| $\tilde{\theta}_k(t) \in \mathbb{R}^D$ | normalized local model of client $k$ at round $t$ |
| $\mathcal{L}(\theta)$ | global loss |
| $\mathcal{L}_k(\theta)$ | client loss |
| $\mathcal{P}_t$ | subset of participating clients at round $t$ |
| $\theta_{\mathrm{gl}}^d \in \mathbb{R}^K$ | random vector of $d$-th components of the global model |
| $\theta^d \in \mathbb{R}^K$ | random vector of $d$-th components of the model |
| $\bar{\theta}^d$ | mean of random vector $\theta^d$ |
| $\bar{\theta}_k(t+1)$ | mean of unobserved clients $k \in [K] \backslash \mathcal{C}_i(t)$ |
| $C^d$ | covariance of random vector $\theta^d$ |
| $\rho_{jk}$ | Pearson correlation coefficient between random variables $j$ and $k$ |
| $v^d \in \mathbb{R}^K$ | value vector for the $d$-th component; $v_k^d$ is the variance reduction for estimating $\theta_{\mathrm{gl}}^d$ when sampling $\theta_k^d$ |
| $v_k$ | total variance reduction associated with client $k$ |
| $\Pi(K)$ | set of all partitions of $[K]$ |
| $\mathcal{C}_i(t) \in \Pi(K)$ | $i$-th coalition at round $t$ |
| $\pi_p(k; t)$ | Boltzmann-like probability that client $k$ is selected in coalition $\mathcal{C}_i(t)$ |
| $n_k$ | number of training samples of $k$-th client |

Table 3: Paper notation summary.

## A  NOTATION

For readability, we summarize below the notation conventions adopted throughout the paper.

## B  THEORETICAL GUARANTEES: PROOFS AND DETAILS

This appendix provides a comprehensive overview of the theoretical framework behind `FedCVR-Bolt`. In Section B.1 we introduce the analogy between social dynamics and FL, focusing on the construction of the influence matrix $W$ between clients' models. In Section B.2, we detail the theoretical foundation of our algorithm, providing formalism and rigor to our design choices. Finally, in Section B.3 we motivate the update policy for each client's mean given the sampled and observed clients.

### B.1  SOCIAL DYNAMICS SETTING

In the last decades, the analysis of Social Dynamics has gained significant attention in the scientific community, providing tools to understand, predict, and influence the spread and dissemination of information across a complex network as well as collective decision-making processes. The application of Opinion Dynamics goes beyond this specific application spacing among marketing, politics, decision-making strategies (Zha et al., 2021). In order to face classical challenges from Federated Learning, we decide to interpret them through the lens of Social Dynamics. Such an innovative angle allows us to establish new results and gain novel insights on the topic.

For the sake of completeness, we briefly present the general framework of Opinion Dynamics on networks. Consider a set of nodes $\mathcal{V}$ representing social individuals in a network. Each node $k \in \mathcal{V}$ is associated with a dynamic state $x_k(t) \in \mathbb{R}$, representing its opinion on a given topic at time $t$. This opinion evolves over time, capturing the opinion formation process. Specifically, at each time step $t$, each agent updates its opinion based on a linear combination of its own opinion, the opinions of others with whom $k$ interacts, and possibly some external input.

Nodes interact on a network. A directed link from node $j$ to node $k$, denoted as $(k, j)$ indicates that agent $j$ influences agent $k$. The set of all directed links is denoted as $\mathcal{E}$. The influence strength among individuals is encoded in a weight matrix $W$, whereby the generic entry $W_{kj}$ represents the influence strength of agent $j$ on agent $k$. Clearly, if $(k, j) \notin \mathcal{E}$, then necessarily $W_{kj} = 0$, since $k$ is not influenced by $j$. The matrix $W$ can be either static or time-varying. In the latter case, the network is commonly referred to as a temporal network. A classical example of time-varying influence matrix is the one used in bounded-confidence models, whereby two agents interact if and only if the difference of their opinions is lower than a given threshold. In wider terms this kind of models aims to capture homophily tendency among agents, which is a peculiar characteristic of many social networks.

In the case of interest, we may interpret each client $k$ as a network node while the node status refers to the client local model $\theta_k(t)$ with $t$ index of the communication round. To naturally model the influence matrix $W(t)$ while accounting for the homophily effect, we build upon the approach introduced in Mäs et al. (2010). Specifically, we define the social influence of agent $j$ on agent $k$ as

$$W_{kj}(t) = \frac{e^{-\gamma\|\theta_k(t)-\theta_j(t)\|_2^2}}{\sum_{m\in\mathcal{K}} e^{-\gamma\|\theta_k(t)-\theta_m(t)\|_2^2}} \quad \forall j, k \in \mathcal{K},$$

where $\gamma > 0$ is a parameter that amplifies the role of models similarities, *i.e.,* the larger $\gamma$, the more pronounced is the homophily effect.

**Remark A1.** *It is worth noting that we expect clients belonging to the same cluster to converge toward similar local models. Consequently, the influence matrix $W$ will asymptotically exhibit a block-like structure, with each block corresponding to a distinct cluster. This observation supports the specific formulation of $W$, as it aligns with our objective of accurately identifying and distinguishing clusters within the network.*

To validate our choice for the proximity matrix, we conduct an ablation study to analyze our framework's robustness. We implement our algorithm with other similarity functions used in coalition formation (Scholkopf & Smola, 2001):

- **Cosine Similarity**: $W_{kj} = \frac{\langle\theta_k(t), \theta_j(t)\rangle}{\|\theta_k(t)\|_2 \cdot \|\theta_j(t)\|_2}$
- **Laplacian Kernel**: $W_{kj} = \exp(-\gamma\|\theta_k(t) - \theta_j(t)\|_1)$
- **Sigmoid Kernel**: $W_{kj} = \tanh(\gamma\langle\theta_k(t), \theta_j(t)\rangle + c)$

While our theoretically-motivated choice yields the best performance (90.23% on MNIST, 57.06% on CIFAR-10), our framework continues to substantially outperform the baselines even with these alternative similarity metrics.

Table 4: Test accuracy (%) across proxy metrics. Best per dataset in bold.

| Proxy Metric | RBF (ours) | Cosine Similarity | Laplacian Kernel | Sigmoid Kernel |
|---|---|---|---|---|
| **MNIST** | **90.23%** | 88.72% | 88.28% | 88.92% |
| **CIFAR-10** | **57.06%** | 55.32% | 49.85% | 50.25% |

## B.2 Sampling Strategy

Once the clusters have been identified, our goal is to determine a representative set of clients to sample, one for each cluster. The approach we adopt is a generalization of classical methods originally developed for sensor placement (Das & Kempe, 2008), and later extended to the context of opinion dynamics in Raineri et al. (2025).

**Assumption A1.** *Given the $K$ random variables $\theta_k$, we indicate with $C^d$, for each $d$-th model component, the covariance matrix $\mathbb{C}ov(\theta^d)$. We assume $C^d$ to be invertible.*

**Remark A2.** *This assumption is not very restrictive, as a violation would imply that the matrix $C^d$ is not full rank—i.e., there exist two clients whose models are identical or linearly dependent. Such cases may arise, for example, in sibling attacks, where one client replicates the data of another. However, exact model equivalence would require not only identical data but also identical training dynamics, including the same mini-batch sampling during stochastic gradient descent. This is highly unlikely in practice.*

*Moreover, our focus is on heterogeneous federated learning under non-adversarial conditions; handling Byzantine or malicious behaviors is beyond the scope of this work. Therefore, we make the standard assumption that the probability of two clients producing exactly identical models is equal to zero.*

Given $\theta_{gl}$ the global model to estimate from a subset $\mathcal{A} \subseteq \mathcal{K}$ of local model sampling, we choose as metrics to quantify the quality of the estimate the **Variance Reduction**, which measures the reduction in uncertainty about $\theta_{gl}$ conditioned to the selected subset $\mathcal{A}$, *i.e.,*

$$v_{\mathcal{A}}^d := \mathbb{V}ar(\theta_{gl}^d) - \mathbb{E}[\mathbb{V}ar(\theta_{gl}^d|\theta_{\mathcal{A}}^d)].$$

For the sake of completeness, let us now provide the technical proofs of the main results presented in Section 3.1 which leads to a proper mathematical formulation of the measure of interest.

For the sake of clarity in our exposition, we prove the statements under the assumption of a zero mean distribution of $\theta^d$. It should be noted that this assumption does not limit the generality of our findings, given that the measure of interest $v_{\mathcal{A}}^d$ is translation invariant, as indicated in *i.e.,*

$$v_{\mathcal{A}}^d = \mathbb{V}ar(\theta_{gl}^d) - \mathbb{E}[\mathbb{V}ar(\theta_{gl}^d|\theta_{\mathcal{A}}^d)] = \mathbb{V}ar(\theta_{gl}^d - \bar{\theta}_{gl}^d) - \mathbb{E}[\mathbb{V}ar(\theta_{gl}^d - \bar{\theta}_{gl}^d|\theta_{\mathcal{A}}^d - \bar{\theta}_{\mathcal{A}}^d)].$$

**Lemma A1.** *Consider $\theta^d \in \mathbb{R}^K$ the zero mean vector containing all the $d$-th model components of the federation and $\theta_{gl}^d = \alpha^\top \theta^d$ the global model to estimate. Given $\mathcal{A} \subseteq K$, it holds*

$$\mathbb{E}[\theta_{gl}^d|\theta_{\mathcal{A}}^d] = (C_{\mathcal{A}\mathcal{A}}^d)^{-1}(C^d\alpha)_{\mathcal{A}}\theta_{\mathcal{A}}^d, \tag{13}$$

*where $C_{\mathcal{A}\mathcal{A}}^d$ is the invertible covariance matrix of $\theta_{\mathcal{A}}^d$.*

*Proof.* Recalling federated averaging aggregation (McMahan et al., 2017), at each round the global model is updated as a linear convex combination of local models (refer to Eq.equation 2). Furthermore, from classical statistical literature (Section 2.3.1 in Hastie et al. (2009)) it is well known that the linear least squares projection of the global model $\theta_{gl}^d$ given the sampling $\theta_{\mathcal{A}}^d$ is computed as

$$\mathbb{E}[\theta_{gl}^d|\theta_{\mathcal{A}}^d] = \hat{\alpha}^\top \theta^d,$$

with

$$\hat{\alpha} = \arg\min_{\beta:\text{supp}(\beta)\subseteq\mathcal{A}} \mathbb{E}[(\theta_{gl}^d - \beta^\top \theta^d)^2],$$

which is the best linear predictor (in the mean squared error sense) of $\theta_{gl}^d$ using only the components of $\theta^d$ belonging to $\mathcal{A}$.

Using the fact that by definition in equation 2 it holds $\theta_{gl}^d = \alpha^\top \theta^d$, we now derive the following series of equalities:

$$\begin{aligned}
\hat{\alpha} &= \arg\min_{\beta:\text{supp}(\beta)\subseteq\mathcal{A}} \mathbb{E}[(\alpha^\top \theta^d - \beta^\top \theta^d)^2] \\
&= \arg\min_{\beta:\text{supp}(\beta)\subseteq\mathcal{A}} \mathbb{E}[(\alpha - \beta)^\top \theta^d(\theta^d)^\top(\alpha - \beta)] \\
&= \arg\min_{\beta:\text{supp}(\beta)\subseteq\mathcal{A}} (\alpha - \beta)^\top C^d(\alpha - \beta) \\
&= \arg\min_{\beta:\text{supp}(\beta)\subseteq\mathcal{A}} -2\beta^\top C^d\alpha + \beta^\top C^d\beta,
\end{aligned}$$

The third equality is derived in accordance with $\mathbb{E}[\theta^d(\theta^d)^\top] = C^d$, given that $\theta^d$ is predicated on the assumption of being zero mean.

In order to compute the minimum let us now impose that

$$\frac{\partial(-2\hat{\alpha}^\top C^d\alpha + \hat{\alpha}^\top C^d\hat{\alpha})}{\partial\hat{\alpha}_k} = 0,$$

for every $k \in \mathcal{A}$, which coincides with

$$(C^d \alpha)_{\mathcal{A}} - (C^d \hat{\alpha})_{\mathcal{A}} = 0 \, .$$

Thus, it holds

$$C^d_{\mathcal{A}\mathcal{A}}(\alpha_{\mathcal{A}} - \hat{\alpha}_{\mathcal{A}}) + C_{\mathcal{A}-\mathcal{A}}\alpha_{-\mathcal{A}} = 0,$$

where $C^d_{\mathcal{A}-\mathcal{A}} = \{C^d_{ij}\}_{i \in \mathcal{A}, j \notin \mathcal{A}}$ and $\alpha_{-\mathcal{A}} = \{\alpha_i\}_{i \notin \mathcal{A}}$.

Given now that $C^d$ is positive definite since it is an invertible covariance matrix, then from Sylvester Criterion also its submatrix $C^d_{\mathcal{A}\mathcal{A}}$ is invertible and the thesis follows. $\qquad\square$

Building on this Lemma, we can now prove the main result used in the definition of the sampling strategy of our algorithm. In other words, let us now prove the explicit formulation for the variance reduction formula.

**Proposition A1.** *Consider $\theta^d \in \mathbb{R}^K$ the zero mean vector containing all the d-th model components of the federation and $\theta^d_{gl} = \alpha^\top \theta^d$ the global model to estimate. Then, the variance reduction, subjected to sampling the subset $\mathcal{A}$ of possible clients, is computed as*

$$v^d_{\mathcal{A}} = (C^d \alpha)^\top_{\mathcal{A}} (C^d_{\mathcal{A}\mathcal{A}})^{-1} (C^d \alpha)_{\mathcal{A}}, \qquad (14)$$

*where $C^d$ is the covariance matrix which captured the correlations among the clients.*

*Proof.* First, notice that from Law of Total Variances

$$\mathbb{V}ar(\theta^d_{gl}) - \mathbb{E}[\mathbb{V}ar(\theta^d_{gl}|\theta^d_{\mathcal{A}})] = \mathbb{V}ar(\mathbb{E}[\theta^d_{gl}|\theta^d_{\mathcal{A}}]) \, .$$

Thus, based on $v^d_{\mathcal{A}}$ definition from equation 12, applying Lemma 1, defined $\hat{\alpha}$ such that

$$\hat{\alpha}_{\mathcal{A}} = (C^d_{\mathcal{A}\mathcal{A}})^{-1}(C^d \alpha)_{\mathcal{A}} \quad , \quad \hat{\alpha}_{-\mathcal{A}} = 0,$$

and known from the steps in the proof of previous Lemma

$$(C^d \alpha)_{\mathcal{A}} = (C^d \hat{\alpha})_{\mathcal{A}},$$

then it holds true that

$$v^d_{\mathcal{A}} = \mathbb{V}ar(\mathbb{E}[\theta^d_{gl}|\theta^d_{\mathcal{A}}]) = \hat{\alpha}' C^d \hat{\alpha} = \hat{\alpha}'_{\mathcal{A}} (C^d \hat{\alpha})_{\mathcal{A}} = \hat{\alpha}'_{\mathcal{A}} (C^d \alpha)_{\mathcal{A}} = (C^d \alpha)'_{\mathcal{A}} (C^d_{\mathcal{A}\mathcal{A}})^{-1} (C^d \alpha)_{\mathcal{A}} \, .$$

$\square$

Despite this metrics allows us to find the best subset of models to sample in order to best predict the global model $\theta_{gl}$, from Raineri et al. (2025) it is known to be a combinatorial problem which becomes unfeasible increasing the number of samples.

Let us observe that if only one observation is allowed then the variance reduction formula, as stated in Corollary 1 in Section 3.1, given $\mathcal{A} = \{k\}$, takes the following form for every $k \in \mathcal{K}$:

$$v^d_k = \frac{(C^d \alpha)^2_k}{C^d_{kk}},$$

which requires $O(K^2)$ operations, significantly reducing the computational cost.

Finally, our heuristic is based on the observation that if the network consists of $P$ connected components, interpreted as coalitions formed through agents' homophily-driven interactions, then a natural sampling strategy is to select one representative client per coalition. This choice is motivated by the goal of optimally reducing the overall variance when the sampling budget allows for $P$ clients.

### B.3 UPDATE POLICY

Once the clients have been selected and sampled, the average of the random variables associated to the unobserved models are updated conditioned to the observations done. An explicit formulation of the conditional expected value is computed in the following proposition.

**Proposition A2.** *Let $\theta^d \in \mathbb{R}^K$ be the vector containing all the $d$-th model components of the federation and $\theta_{gl}^d = \alpha^\top \theta^d$ the global model to estimate. Let $\theta_k^d$ be the sampled model at round $t$. Then, the optimal estimator of an arbitrary model $d$-th component $\theta_j^d$ with $j \neq k$ is*

$$\mathbb{E}[\theta_j^d | \theta_k^d] = \rho_{kj} \frac{\sqrt{v_j}}{\sqrt{v_k}} \theta_k^d, \tag{15}$$

*with $\rho_{kj}$ Pearson Correlation Coefficient between variables $k$ and $j$.*

*Proof.* First, notice that, from Lemma 1, chosen $\alpha = \delta^{(j)}$ where $\delta_i^{(j)} = \begin{cases} 1 & i = j \\ 0 & i \neq j \end{cases}$, it holds

$$\mathbb{E}[\theta_j^d | \theta_k^d] = \frac{C_{kj}^d}{C_{kk}^d} \theta_k^d.$$

Recalling that by $\rho$ definition it holds

$$\rho_{jk} = \frac{C_{kj}^d}{\sqrt{C_{kk}^d} \sqrt{C_{jj}^d}},$$

the thesis follows. $\qquad\square$

### B.4 CONVERGENCE ANALYSIS

In this appendix, we establish convergence guarantees for `FedCVR-Bolt`. Our analysis builds on standard assumptions in FL – smoothness of the global loss and bounded variance of the stochastic federated gradient – extended with a mild alignment condition on the client selection policy. The latter ensures that, in expectation, the update direction preserves a positive correlation with the true gradient, thereby maintaining descent. Under these assumptions, we show that the sequence of global iterates produced by `FedCVR-Bolt` converges, in expectation, to a neighborhood of a stationary point of the global loss. Here, convergence is meant in the stochastic optimization sense: once averaged over both the randomness of client sampling and the evolution in time, the gradient norm becomes small. This differs from the stronger notion of convergence of the iterates $\theta_{gl}$ themselves to an exact stationary point, which is precluded by the intrinsic variance of stochastic updates but replaced by concentration around regions of vanishing gradient.

**Assumption A2.** *The global loss function $\mathcal{L}(\theta)$ is $L$-smooth, i.e., it is differentiable with $L$-Lipschitz gradient and its federated gradient $g(t) := \sum_{k \in P_t} \tilde{\alpha}_k \nabla \mathcal{L}_k(\theta_{gl}(t))$ has bounded variance, i.e., $\mathbb{E}_{\mathcal{P}_t \sim \pi(t)}[\|g(t) - \mathbb{E}[g(t)]\|^2] < \sigma^2$.*

**Assumption A3.** *Gradient Variance Alignment. Let $G(\theta) := \mathbb{E}_{\mathcal{P}_t \sim \pi(t)}[g(t)]$. Assume there exists a constant $c > 0$ s.t. $\langle G(\theta), \nabla \mathcal{L}(\theta) \rangle \geq c \|\nabla \mathcal{L}(\theta)\|^2$, i.e., the bias of the policy is informed and points in a descent direction (very weak, and can be verified by our experimental results).*

**Proposition A3.** *Let $\{\theta_{gl}(t)\}_t$ the sequence of global model update produced by FedCVR-Bolt algorithm, with known model covariance $C$. Under Assumptions 1 and 2, FedCVR-Bolt, for a small learning rate $\eta > 0$, converges to a neighborhood of a stationary point, i.e.,*

$$\lim_{T \to \infty} \frac{1}{T} \sum_{t=0}^{T-1} \mathbb{E}[\|\nabla \mathcal{L}(\theta_{gl})\|^2] < \varepsilon.$$

*Proof.* We focus on a single SGD step, having $\theta_{gl}(t+1) = \theta_{gl}(t) - \eta g(t)$, where $g(t)$ is the federated gradient produced by FedCVR-Bolt. Due to Assumption 1, if the loss is $L$-smooth, we can apply the Descent Lemma, from Chapter 2, Theorem 2.1.5 in Nesterov (2014). Thus, we bound as follows:

$$\mathcal{L}(\theta_{gl}(t+1)) \leq \mathcal{L}(\theta_{gl}(t)) - \eta \langle \nabla \mathcal{L}(\theta_{gl}(t)), g(t) \rangle + \frac{L\eta^2}{2\|g(t)\|^2}.$$

By taking the expectation up to the $t$-th round, denoted by $\mathbb{E}_t$, our bound becomes:

$$\mathbb{E}_t[\mathcal{L}(\theta_{gl}(t+1))] \leq \mathcal{L}(\theta_{gl}(t)) - \eta \langle \nabla \mathcal{L}(\theta_{gl}(t)), \mathbb{E}_t[g(t)] \rangle + \frac{L\eta^2}{2\mathbb{E}_t[\|g(t)\|^2]}$$

Without losing generality, we can assume that after $T_0$ we are in a thermalized regime, where the spectral clustering has converged, and the clusters are stationary, namely $\forall t > T_0$, $\mathcal{C}_p(t) = \mathcal{C}_p^*$, hence also the Boltzmann measure $\pi(t)$ is stationary. Let us recall that we defined $\mathbb{E}_t[g(t)] = G(\theta_{gl}(t))$. Applying Assumption 2, we establish the following bounds:

$$\langle \nabla \mathcal{L}(\theta_{gl}(t)), \mathbb{E}_t[g(t)] \rangle = \langle \nabla \mathcal{L}(\theta_{gl}(t)), G(\theta_{gl}(t)) \rangle \geq c \|\nabla \mathcal{L}(\theta_{gl}(t))\|^2,$$

and

$$\mathbb{E}_t[\|g(t)\|^2] \leq \|G(\theta_{gl}(t))\|^2 + \mathbb{E}_t[\|g(t) - G(\theta_{gl}(t))\|^2] \leq \|G(\theta_{gl}(t))\|^2 + \sigma^2 =: M,$$

since the gradient updates have bounded variance.

Therefore, we can re-write the bound obtained in the above as

$$\mathbb{E}_t[\mathcal{L}(\theta_{gl}(t+1))] \leq \mathcal{L}(\theta_{gl}(t)) - \eta c \|\nabla \mathcal{L}(\theta_{gl}(t))\|^2 + L\eta^2 M/2$$

By taking the expectation over the whole iterations, denoted by $\mathbb{E}$, we get

$$\mathbb{E}[\mathcal{L}(\theta_{gl}(t+1))] \leq \mathbb{E}[\mathcal{L}(\theta_{gl}(t))] - \eta c \mathbb{E}[\|\nabla \mathcal{L}(\theta_{gl}(t))\|^2] + L\eta^2 M/2$$

If we sum over times, from the termalization time $T_0$ to the final round $T - 1$, and we rearrange the inequalities, we get

$$\eta c \sum_{t=T_0}^{T-1} \mathbb{E}[\|\nabla \mathcal{L}(\theta_{gl}(t))\|^2] \leq \sum_{t=T_0}^{T-1} (\mathbb{E}[\mathcal{L}(\theta_{gl}(t))] - \mathbb{E}[\mathcal{L}(\theta_{gl}(t+1))]) + (T - T_0)L\eta^2 M/2$$

Expanding the telescoping sum term on the right-hand-side, it reduces to $\mathbb{E}[\mathcal{L}(\theta_{gl}(T_0))] - \mathbb{E}[\mathcal{L}(\theta_{gl}(T))]$. Let $\mathcal{L}^*$ be the global optimum of the global loss, then $[\mathcal{L}(\theta_{gl}(T)] \geq \mathcal{L}^*$ and the expression above becomes

$$\eta c \sum_{t=T_0}^{T-1} \mathbb{E}[\|\nabla \mathcal{L}(\theta_{gl}(t))\|^2] \leq \mathcal{L}(\theta_{gl}(T_0)) - \mathcal{L}^* + (T - T_0)L\eta^2 M/2$$

Dividing both terms by $\eta c (T - T_0)$ the inequality becomes

$$\frac{1}{T - T_0} \sum_{t=T_0}^{T-1} \mathbb{E}[\|\nabla \mathcal{L}(\theta_{gl}(t))\|^2] \leq \frac{\mathcal{L}(\theta_{gl}(T_0))}{\eta c (T - T_0)} - \frac{\mathcal{L}^*}{\eta c (T - T_0)} + \frac{L\eta M}{2c}$$

For $T \to \infty$, $T_0$ is negligible with respect to $T$. Hence, by finally adopting a change of variable and taking the limit with respect to $T$, we obtain the claim as

$$\lim_{T \to \infty} \frac{1}{T} \sum_{t=0}^{T} \mathbb{E}[\|\nabla \mathcal{L}(\theta_{gl}(t))\|^2] \leq \frac{L\eta M}{2c} =: \varepsilon$$

$\square$

# C IMPLEMENTATION DETAILS AND FURTHER EXPERIMENTS

In this appendix, we focus on the experimental setup, hyperparameters of `FedCVR-Bolt`, and baselines, including datasets and models. Moreover, Section C.2 addresses the practical implementation of `FedCVR-Bolt` in the context of increasing model dimensions. Finally, Section C.3 provides a detailed analysis of the synthetic dataset structure that simulates synthetic heterogeneous linear regression for FL.

Code is available at `https://anonymous.4open.science/r/fedcvr_bolt-26C8`.

## C.1 DATASETS AND IMPLEMENTATION DETAILS

We evaluate `FedCVR-Bolt` on various datasets. For a more straightforward scenario, we consider a one-dimensional linear regression under both IID and non-IID distributions, with and without intercept, *i.e.,* $D = 1$ and $D = 2$ respectively. The dataset construction process is outlined in Section C.3. Subsequently, we evaluate the performance of `FedCVR-Bolt` using several classification benchmark datasets that are widely used in FL (Li et al., 2020; Caldas et al., 2018), particularly utilizing the Synthetic dataset (Li et al., 2020; Cho et al., 2022), Federated MNIST (LeCun, 1998), and CIFAR-10 (Krizhevsky, 2009). To simulate a heterogeneous environment, we employ the Synthetic(1,1) setting, while the other datasets are partitioned using a Dirichlet distribution with $\alpha = 0.1$, by implementing a sampler according to Caldas et al. (2018). Regarding the MNIST dataset, a multilayer perceptron was utilized, comprising two layers with 200 hidden neurons and ReLU activations. In contrast, the CIFAR-10 and CIFAR-100 experiments involved a convolutional neural network, which consisted of two convolutional and max-pooling layers, followed by two fully connected layers, for generating class predictions.

In all experiments, the dataset is partitioned across $K = 100$ clients, followed by dividing the datasets into training and testing sets to assess model performance. The models have been trained with Stochastic Gradient Descent with $S = 10$ local epochs, a batch size of 100, and a learning rate of $\eta = 0.01$. We set the number of communication rounds to $T = 200$ for the MNIST dataset experiments and $T = 100$ for the other datasets. For each round, $P = 10$ clients are sampled to participate in the training.

We employed $\gamma_t = 1/t$ in the training of `FedCVR-Bolt` to ensure the convergence of Robbins-Monro estimators applied to the covariance (Robbins & Monro, 1951), followed by the temperature parameter $\beta = 1$. In the case of `Power-of-choice` (Cho et al., 2022), we obtained $d = 2P$ samples and subsequently selected the $P$ that exhibited the highest test loss on the global model. For `Active FL` (Goetz et al., 2019), we set $\alpha_1 = 0.8$ (which is equivalent to $d = 2P$), and similarly utilized the temperature parameter $\alpha_2 = 1$ as our temperature parameter $\beta$, $\alpha_3 = 0$ to achieve comparability with `Power-of-choice`.

Synthetic experiments were executed locally on an Apple M1 processor, while MNIST and CIFAR experiments utilized an `RTX8000 NVIDIA` GPUs.

## C.2 PRACTICAL IMPLEMENTATION OF FEDCVR-BOLT

Practically, `FedCVR-Bolt` clusters client according to the observed model $\theta_k(t)$. To further stabilize the initial estimation phase, we implement a uniformly random sampling strategy during the initial 30 rounds, enabling the observation of local adaptations of the global model across various sampled clients. The selection of 30 rounds offers a balanced approach between exploring client heterogeneity and subsequently leveraging the clustering structure. This initial phase is dedicated to observing a diverse set of client behaviors, which allows the Robbins-Monro estimator to build a more reliable and stable baseline covariance structure before our variance-reduction sampling policy becomes active (Grimmett & Stirzaker, 2001).

Similar to numerous FL methods based on local updates — such as Ghosh et al. (2020) — executing operations on the full model becomes increasingly burdensome as the model size increases. Furthermore, from a statistical standpoint, the early feature extraction layers (such as convolutional or embedding layers) frequently contribute less discriminative information for clustering and may introduce superfluous noise along with computational overhead. Hence, when managing larger models, we confine the covariance computation to the parameters of the final fully connected layer. If this layer

remains excessively large, we further diminish dimensionality by randomly sampling a subset of its weights. Specifically, for the MNIST dataset, we utilize the entire final layer; whereas for CIFAR-10, where the final layer has higher dimensionality, we sample $D = 300$ weights. This approach achieves an advantageous compromise between computational efficiency and representational adequacy.

### C.3 SYNTHETIC LINEAR REGRESSION

To simulate client heterogeneity, we generate synthetic datasets across $K = 100$ clients, each assigned to one of $J$ latent clusters. The IID case is reproducible by setting $J = 1$. Each client $k$ is associated with a cluster index $j_k$, sampled uniformly from $0, \ldots, J - 1$. Given $j_k$, we generate local input samples $\{x_k^i\}_{i=1}^N \subset \mathbb{R}^D$ from a Gaussian distribution $\mathcal{D}(\theta_{x,j_k}, \sigma_{x,j_k}^2 I_D)$, where $\theta_{x,j_k}$ is the cluster-dependent mean, *i.e.,* the mean of the input data distribution for cluster $j_k \in \{0, \ldots, J - 1\}$, and $\sigma_{x,j_k}$ is the corresponding standard deviation. For each input $x_k^i$, a latent regression vector $\theta_k^i \sim \mathcal{D}(\bar{\theta}_{j_k}, \sigma_{\theta,j_k}^2 I_D)$ is independently sampled, and the target variable is computed as $y_k^i = (\theta_k^i)^\top x_k^i$. To enable an intercept term, a bias component of 1 is concatenated to each $x_k^i$. This process yields a rich, continuous label space with both intra and inter cluster variability, reflecting both statistical heterogeneity across clients and random variations within each local dataset.

## D PRIVACY OF FedCVR-Bolt

The privacy of client data in FedCVR-Bolt algorithm is preserved by adhering to the fundamental principles of FL: raw data remains on client devices and is never transmitted to the server. Clients selected from the set $\mathcal{P}_t$ perform local updates using the global model $\theta_{gl}(t)$ to produce their updated local models $\theta_{j_p}(t + 1)$. Moreover, clients subsequently communicate to the server only these model parameters, not the underlying private data. To further protect such parameters during aggregation steps (e.g., for the computation of $\theta_{gl}(t + 1)$), it is possible to employ secure aggregation protocols (Bonawitz et al., 2016). While the distinctive server-side operations of FedCVR-Bolt , such as clustering based on individual models $\{\theta_k(t)\}$ and deriving $\bar{\theta}_k(t + 1)$ from specific client model updates $\theta_{j_p}(t + 1)$, require the server to access these individual parameters for its core functionality, this architectural choice is consistent with federated architectures where the central server orchestrates the learning process using the model parameters received from the clients (Kairouz et al., 2021) for advanced tasks like personalization or clustering (Smith et al., 2017). Thus, FedCVR-Bolt ensures foundational data privacy by localizing data processing, and the principles of secure aggregation offer a complementary mechanism for enhancing model confidentiality during aggregation where the algorithm's direct need for individual parameter access permits.

