# OpenReview forum: "Socially Inspired Coalition Formation and Client Selection in Federated Learning"
_ICLR.cc/2026/Conference — ICLR 2026 Conference Withdrawn Submission_

### Official Review · Reviewer_A8ac · 2025-10-30

**Soundness:** 2
**Presentation:** 2
**Contribution:** 2
**Rating:** 4
**Confidence:** 3

**Summary:**

The author propose FedCVR-Bolt , a client sampling algorithm, which improves performance in heterogeneous FL settings compared to existing sampling strategies.

**Strengths:**

1. Innovatively incorporates social dynamics models, treating clients as nodes in a social network and using homophily to construct influence matrices, which is inspiring.
2. heoretical foundation with mathematical derivations for variance reduction and convergence guarantees, enhancing credibility.

**Weaknesses:**

1. Experimental models are too simplistic; only MLP and a simple CNN are used, lacking validation on more complex architectures like ResNet or RegNet.
2. The number of coalitions P is fixed a priori, which may limit adaptability in dynamically changing data environments; sensitivity analysis on P is missing.
3. Insufficient computational cost analysis; no actual runtime or resource consumption metrics are provided.
4. Limited heterogeneity simulation; only Dirichlet distribution with α=0.1 and fixed client count are used, lacking diversity in settings and ablation studies.
5. Baseline comparisons are outdated; no comparison with more recent state-of-the-art methods

**Questions:**

Please see weaknesses

---

### Official Review · Reviewer_ccPy · 2025-11-01

**Soundness:** 3
**Presentation:** 2
**Contribution:** 2
**Rating:** 2
**Confidence:** 5

**Summary:**

This paper introduces FedCVR-Bolt, a novel client selection algorithm for Federated Learning (FL)  designed to mitigate challenges from statistical heterogeneity. The method draws inspiration from social dynamics and involves two stages: (1) Coalition Formation: It uses a homophily-based similarity matrix and spectral clustering  to dynamically group clients into "coalitions." (2) Client Selection: From each coalition, it probabilistically selects one representative client by maximizing variance reduction combined with Boltzmann exploration.

**Strengths:**

1.The introduction of concepts from social dynamics, such as "coalition formation" and "homophily", to the FL client selection problem provides an innovative and intuitive framework for handling client clustering.
2.The algorithm is not just a heuristic. The authors provide detailed theoretical derivations for the variance reduction metric (Propositions 1 & 2) and a convergence analysis (Proposition A3), which strengthens the method's reliability.

**Weaknesses:**

The experimental results are insufficient, and the latest methods need to be compared.

The current classification experiments are limited to an extreme heterogeneity setting (α=0.1) , and the regression experiments are confined to a simple cluster setup (J=2) . To more comprehensively evaluate the algorithm's robustness across different degrees of heterogeneity, the authors can supplementing the experiments with moderate (e.g., α=0.5) or mild (e.g., α=1.0) non-IID settings.

**Questions:**

The method's core relies on estimating, storing, and computing with D separateK ×K covariance matrices (C^d). This introduces a significant O(D×K^2 ) storage overhead on the server. More critically, when the number of clients K is large, and especially when the model dimension D is large (e.g., millions of parameters in modern neural networks), the O(D×K^2 )  computational cost required to calculate all  v_k values in each round becomes prohibitive.

---

### Official Review · Reviewer_qJjp · 2025-11-01

**Soundness:** 2
**Presentation:** 3
**Contribution:** 3
**Rating:** 4
**Confidence:** 4

**Summary:**

This paper studied the problem of federated learning under data heterogeneity. It introduced a client-selection algorithm, which dynamically formed non-overlapping coalitions of clients and selected one representative from each coalition to minimize the variance of model updates. The efficacy of the proposed client-selection algorithm was verified theoretically and empirically.

**Strengths:**

**Originality:** It introduced a novel client sampling algorithm FedCVR-Bolt for heterogeneous federated learning. The Boltzmann exploration policy was adopted to improve the client sampling strategy. Finally, the convergence of the FedCVR-Bolt algorithm was analyzed.

**Quality:** The sampling strategy based on the variance reduction was theoretically justified. Experimental results on the synthetic regression and real-world image datasets demonstrated the effectiveness of the proposed FedCVR-Bolt algorithm.


**Clarity:** The proposed sampling and updating policies were well-motivated. The proposed FedCVR-Bolt was easy to follow.


**Significance:** It provided a novel client sampling method to advance federated learning under data heterogeneity.

**Weaknesses:**

(1) The impact of the Boltzmann exploration policy on the variance reduction is unclear. Subsection 3.1 justified the rationale of the sampling strategy in Eq. (5) based on the variance reduction. The Boltzmann exploration policy transforms the deterministic selection based on the estimated variance into a Boltzmann-like probability measure. However, it is unclear whether this probability measure in Eq. (8) theoretically supports the variance reduction like Corollary 1. Besides, lines 259-260 state that it can "mitigate the risk of becoming trapped by early biased or noisy estimates of the covariance matrix". This can be further justified. Appendix C.2 mentions that a uniformly random sampling strategy during the initial 30 rounds is used. Then why is the Boltzmann exploration policy required to mitigate the risk of becoming trapped by early biased estimates of the covariance matrix?

(2) The coalition formation and updating policy can be further clarified.
-  Subsection 2.1 presents the partial-participation strategy with the randomly selected subset of clients, which is the commonly used policy in FL with a large number of clients. However, this is fundamentally different from the proposed updating policy in Section 3. It carefully selects one representative per cluster. Instead, the coalition formation requires accessing the model parameters from all clients during each training round. This is more similar to cross-silo FL.
- Lines 249-250 show that selecting one representative per cluster mitigates overfitting to any single sub-distribution. It is unclear why this selection strategy can mitigate overfitting to any single sub-distribution.
- Step 18 of Algorithm 1 shows the estimation of the global aggregated parameters. But it is unclear how it will be used.

(3) The experiments can be significantly improved to justify the advantages of the proposed algorithm.
- The convergence of FedCVR-Bolt can be verified in the experiments. Compared to baselines, will FedCVR-Bolt have an improved convergence rate?
- It would be convincing to visualize what clients are selected during training. Appendix C.2 mentions that a uniformly random sampling strategy during the initial 30 rounds is used. This policy can also be validated.
- The running efficiency of FedCVR-Bolt can also be validated. Section 4 shows the computational cost of $O(K^2D)$. This can be very computationally expensive when the number of clients $K$ is very large.

**Questions:**

(1) Why are some results of baseline FL approaches very weak compared to FedAvg, e.g., FedProx on CIFAR-10 and FeSEM on CIFAR-10/CIFAR-100?

(2) Compared to baselines, the running efficiency of FedCVR-Bolt can be further validated.

---

### Official Review · Reviewer_FuJ6 · 2025-11-01

**Soundness:** 3
**Presentation:** 4
**Contribution:** 2
**Rating:** 4
**Confidence:** 4

**Summary:**

This paper developed a client selection method for federated learning. It is derived from a variance reduction technique (Raineri et al., 2025), and it employs an Boltzmann exploration (Eq.(8)) to balance exploration and exploitation. Experiments were conducted on synthetic regression and standard classification datasets (MNIST & CIFAR) to show its effectiveness.

**Strengths:**

1. The writing and presentation are very clear and easy to follow.

2. The algorithm is theoretically motivated.

**Weaknesses:**

1. There are noticeable drawbacks in the theory.

- The paper neglects the changing nature of model parameter distributions over time. Specifically, it is an over-simplification to say that $\theta_k(t)$ is a realization of the random variable mentioned in L159. What is the underlying distribution? Clearly, the parameters are closer to a local minimum at later stages of learning, so the underlying distribution is changing over time. This has noticeable consequences later. For example, using (11) assumes that the model parameter distributions are stationary.
- The selection is based on a greedy strategy, not accounting for the correlations among parameters. L364 focuses on the sum of reduced variances, which is less helpful when the parameters are correlated.

2. Experiments are limited

- Baselines are limited. The algorithm focuses on client selection, but only two of the baselines are selection methods. It would make sense to compare with other selection methods, such as Balakrishnan et al., (2022), Tian et al., (2022) and Chen et al. (2024).
- The regression problem is too simple with only two clusters and at most two-dimensional parameters. What if we have $J=10$ clusters, but the algorithm sets $P=5$ or $P=20$?
- There are no ablation studies on the choices of the hyper-parameters for the classification experiments, such as $P$, $\gamma$ in (4) and $\beta$ in (8).

Ref

- Tian, C., Shi, Z., Li, L. and Xu, C.Z., 2024, July. Ranking-based client imitation selection for efficient federated learning. In *Forty-first International Conference on Machine Learning*.
- Chen, H. and Vikalo, H., 2024. Heterogeneity-guided client sampling: Towards fast and efficient non-iid federated learning. *Advances in Neural Information Processing Systems*, *37*, pp.65525-65561.

Minor comments

- For some reason, the Related Work section has no section number.
- L341: Raineri et al. (2025) has no Lemma 2.

**Questions:**

Please see the weaknesses above.

---

### Note · Authors · 2025-11-19

**Comment:**

Dear AC, PC, and Reviewers,

First of all, we would like to thank you for having handled our submissions and for the feedback provided and for the observations regarding the methodological aspects of the paper. We recognize that carrying out the additional experiments suggested would require more time than is currently available. For this reason, we have decided to respectfully withdraw our submission.

**Withdrawal Confirmation:**

I have read and agree with the venue's withdrawal policy on behalf of myself and my co-authors.